# Particle-Based Imaging Tools Revealing Water Flows in Maize Nodal Vascular Plexus

**DOI:** 10.3390/plants11121533

**Published:** 2022-06-08

**Authors:** Ulyana S. Zubairova, Aleksandra Yu. Kravtsova, Alexander V. Romashchenko, Anastasiia A. Pushkareva, Alexey V. Doroshkov

**Affiliations:** 1Institute of Cytology and Genetics Siberian Branch, Russian Academy of Sciences, 630090 Novosibirsk, Russia; arom@bionet.nsc.ru (A.V.R.); ad@bionet.nsc.ru (A.V.D.); 2Institute of Computational Mathematics and Mathematical Geophysics Siberian Branch, Russian Academy of Sciences, 630090 Novosibirsk, Russia; 3Department of Mechanics and Mathematics, Novosibirsk State University, 630090 Novosibirsk, Russia; a.pushkareva@g.nsu.ru; 4Kutateladze Institute of Thermophysics Siberian Branch, Russian Academy of Sciences, 630090 Novosibirsk, Russia; kravtsova.alya@gmail.com; 5Institute of Fundamental Biology and Biotechnology, Siberian Federal University, 660036 Krasnoyarsk, Russia

**Keywords:** plant 3D imaging, *Zea mays* L., vascular system, internodes, nodal plexus, contrast-enhanced magnetic resonance imaging, laser scanning microscopy, lab-on-a-chip, Particle Image Velocimetry, systems biology

## Abstract

In plants, water flows are the major driving force behind growth and play a crucial role in the life cycle. To study hydrodynamics, methods based on tracking small particles inside water flows attend a special place. Thanks to these tools, it is possible to obtain information about the dynamics of the spatial distribution of the flux characteristics. In this paper, using contrast-enhanced magnetic resonance imaging (MRI), we show that gadolinium chelate, used as an MRI contrast agent, marks the structural characteristics of the xylem bundles of maize stem nodes and internodes. Supplementing MRI data, the high-precision visualization of xylem vessels by laser scanning microscopy was used to reveal the structural and dimensional characteristics of the stem vascular system. In addition, we propose the concept of using prototype “Y-type xylem vascular connection” as a model of the elementary connection of vessels within the vascular system. A Reynolds number could match the microchannel model with the real xylem vessels.

## 1. Introduction

Plants are living hydrodynamic systems functioning by virtue of the movement of water, which is one of the fundamental factors of their homeostasis [1]. The water uptake is performed by the part of the plant vascular system named xylem, which provides a low-resistance pathway for water from soil to atmosphere through roots, stems, and leaves [2,3]. Plants mainly transport water and dissolved minerals through xylem vessels composed of dead lignified cells connected in a complex system [4].

The water transport system in vascular plants is diverse and often structurally complex [5]. In most grasses, the xylem vessels of the stems differ in topology between nodes and internodes. The internodes comprise the long axial bundles of the metaxylem and protoxylem vessels, while within the nodes, they pass through a network of transverse bundles, and the metaxylem and protoxylem vessels are connected by small tracheary elements [6]. The axial and the transverse vessels contacts generate various forms of vascular elements, usually mediated by three-way Y-type connections. Maize also has these features of vascular system topology [6,7,8]. It is notable that only about 3% of axial vessels pass through nodes straightly [7].

Imaging-based plant physiology study approaches provide an effective tool for many current and challenging tasks that require the precise quantification of objects based on the analysis of their various features. Tools for imaging water flows inside plant tissues are diverse, for example, recent advances in the Raman imaging of water-transporting xylem vessels [9] or the fast neutron tomography of root water uptake [10]. Among them, X-ray micro computed tomography (micro-CT) allows three-dimensional visualization to be obtained, revealing the spatial configuration and distribution of vascular bundles. For example, for the maize stem, the studies [11,12,13,14] obtained such visualization. The structural characteristics of xylem vessels in the vascular bundles of excised maize leaves with high spatial resolution were obtained in [15]. In the study [16], the images of root–soil systems obtained by micro-CT were analyzed to measure the porosity and the displacement fields in the soil near the maize root surface. A combination of micro-CT with the other imaging methods, including confocal microscopy, revealed the vascular architecture of the cauline system in Commelinaceae [17].

Besides the reconstruction of the vascular 3D architecture in various plant organs, the spatio-temporal sap flow characteristics could clarify plenty of important aspects in plant physiology. In this sense, magnetic resonance imaging (MRI), visualizing water protons directly, is a splendid technique for solving outstanding issues in plant science [18]. So, a series of studies presented MRI measurements for phloem transport and xylem transport in poplar, castor bean, tomato and tobacco [19], in woody lianas stem [20], in the tomato peduncle [21], and in a cucumber stem [22]. Note that the power of the magnetic field, varying from 0.47 T to 4.7 T in the listed studies, affects the resulting image quality. In addition, most MRI devices are horizontally oriented, so special modifications are used to study plants. A combination of MRI with positron emission tomography (PET) modified for the study of vertical objects provided new insights into structural and functional traits considering in vivo water and photoassimilate transport [23].

The xylem pathways for water in maize stems were investigated with dye and particulate tracers [7]. A similar idea the particles flow is realized in the MRI contrast-enhanced approach. Paramagnetic compounds acting as a contrast can influence the $T_{1}$ or $T_{2}$ proton relaxation rate. Examples of such compounds are gadolinium chelates [24,25] and manganese salts. The relaxation behavior is associated with interactions between water and other cellular components [26]. Thus, different $T_{2}$ values reflect the different state of water in the materials, and higher $T_{2}$ values represent more mobile water, while lower values indicate that the water molecules are less mobile. The study [27] investigated the state of water in apple parenchyma tissue using paramagnetic ions (divalent manganese ions) and identified three water fractions in different cellular compartments: vacuole, cytoplasm, and cell wall/extracellular space. The study [28] observed three water fractions in broccoli and illustrated different interactions between the three water components and the materials within broccoli tissues.

Still, the existing in vivo imaging tools for fluid transport do not reproduce accurate quantitative data on the spatial interactions of the flows within vessels, while microfluidic devices could simulate transport processes in plants. Therefore, developing lab-on-a-chip methods makes it possible to reproduce the dynamics of the mechanism of liquid movement in plant vessels with wide variation in sizes and connection types. A synthetic tree-on-a-chip representing passive phloem loading and long-distance transport was presented in [29]. An artificial xylem chip representing a 3D-printed vertical digital microfluidic platform was elaborated in [30]. Passive water ascent in a tall, scalable synthetic tree was shown in [31]. An artificial leaf developed with a fluid pump driven by surface tension and evaporation was presented in [32]. Besides that, computational fluid dynamic methods could quantify the internal flow variables of xylem vessels [33].

In this study, we aimed to show a combination of particle-based imaging tools for water flow visualization in a case study of the maize stem vascular system. We used combined contrast-enhanced MRI, confocal laser microscopy, and a microfluidic chip simulating an element of xylem vessel connection and indicated a way to achieve the matching between the xylem vessels in the maize stem and lab-on-a-chip microchannels. Owing to the fundamental possibility of using a horizontal tomograph with a magnetic field strength of 11.7 T as an imaging tool for plants, we determined the difference in the spatial distribution of the water flow characteristics between the internode and the nodal plexus in the maize stem. The presented tools are steps towards a multi-level approach to the biophysical modeling of the transport system.

## 2. Results

### 2.1. MRI Approaches for Water Visualization inside the Xylem Vascular Bundle System

Magnetic resonance imaging (MRI) is an attractive technique providing detailed, non-invasive, and quantitative information about water transport and water balance within the plant vascular system. On the stems of 8-week-old maize plants, we tested ultrahigh-field tomograph BioSpec 117/16 USR (Bruker BioSpin, Ettlingen, Germany) as a tool for obtaining high-resolution images of water transport system structure and functioning. For a shoot region constructed by nested leaves, a set of images for different MRI modes was obtained. On a cross section, tissues with contrast water content were well visualized (Figure 1a–f).

Subtraction the $T_{1}$ from the $T_{2}$-weighted images of the same object made it possible to obtain the proton-weighted (PD) maps for the distribution of aqueous and non-aqueous protons (polysaccharides, glycoproteins) for the maize stem in Figure 1a–c. In addition, images representing the diffusivity (Figure 1d) and flux could be obtained (Figure 1e).

Thus, for the vascular tissues of plants, horizontal tomograph Biospec 117/16 USR allowed functional visualization to be obtained, which represented a combination of anatomical and functional information. The use of $T_{1}$- and $T_{2}$-weighted sequences in comparison with diffusion- and proton-weighted methods made it possible to achieve a higher resolution for the visualization of the conducting system (Figure 1). The accumulation of Mn${}^{2+}$ in the plant tissue affected relaxation time $T_{1}$ to a greater extent than $T_{2}$ (Figure 1f).

With $T_{2}$-weighted sequences, we visualized the structures of the conducting system for the maize stem (Figure 1g–l and Appendix A). On the longitudinal MRI section (Figure 1g), the vascular bundles of the stem, individual leaves, and nodes were distinguishable. It was interesting to note the good visualization of the highly contrastive topology of the vascular bundles in the nodal plexuses (nps) and internodes (ins). The internodes contained parallel vessels without visible contacts between them, whereas inside the nodes, there were multiple vessel interconnections distributing water flows between the leaf of the corresponding stage and the distal part of the stem. Thus, each node contained elements of local topology complication, including the multiple branching and merging of the conducting bundles of the vascular system. Despite the overall complexity of the system, a Y-type compound could be considered as an elementary unit.

Considering the conducted experimental imaging, to obtain information on the dynamics of water distribution in plant organs, we considered it promising to combine high-resolution $T_{2}$-weighted MRI and contrast-enhanced $T_{1}$-weighted MRI. As contrast agents, we used gadolinium chelate (see Section 2.2), which is not able to penetrate into cells and is located in the extracellular space, as well as manganese chloride, whose ions penetrate into the intracellular space through transporter proteins. Based on these data, it was possible to map the distribution of water in intra- and extracellular compartments.

### 2.2. Water Flows in Internodes and Nodal Plexuses Derived from Contrast-Enhanced MRI

Contrast-enhanced MRI may serve as a tool for imaging spatio-temporal water distribution features inside living tissues using contrast agents as flow-markers. As a contrast agent, we used gadolinium chelate, which, once introduced into the xylem vessel, cannot leave it because of its relatively large size. Note that in the nodal plexuses, there are many narrow connections and locked vessels where water passes by the transmembrane pathway. Internodes, on the contrary, comprise xylem vessels with a relatively large lumen. Therefore, this experimental design allowed us to monitor the gadolinium chelate accumulation dynamics in the nodal plexus in vivo during an extended period.

In the experiment, we used a 4-week-old maize plant with a small internode between the root–shoot junction and the following plexus. The MRI-studied area was located at the bottom of the stem, from the junction of the shoot and roots to a height of 18 mm, covering the bottom of the following node with shoot apical meristem and growing leaves (Figure 2a). After the roots were exposed to the contrast agent solution, scanning was performed every 23 min for 7 h. This resulted in the image series of spatio-temporal signal distribution inside the stem (Figure 2b and Appendix A). Although the resolution of this imaging method allows to identify only large vessels, the difference in the distribution pattern for the nodal plexus (slices 1, 5, 17) and internodal (slice 9) regions is clear. On the other hand, laser scanning microscopy (LSM) provided the high-resolution anatomical characteristics of xylem vessels inside vascular bundles (Figure 2c and Appendix A), complementing MRI. The high-precision LSM-visualization of xylem vascular bundles was used to reveal the structural and dimensional characteristics of the vessels, in particular for calculating the hydraulic diameters ($D_{h}$; see Section 2.4).

The spatio-temporal distribution of the signal averaged over the stem clearly shows areas of internodes that did not retain contrast agents, and areas of nodal plexuses that filtered them, thereby accumulating the signal (Figure 2c). The studies [7,8] investigated perfusing maize xylem vascular structures with dye solutions, suspensions of latex particles, and gold sols. The authors showed that gold particles about 5 nm in size (approximately equivalent to gadolinium chelate) accumulated in cell walls along the water pathway. Since water passing through the nodal plexuses is partially transported along the apoplastic pathway through the cell walls, gadolinium chelate could accumulate in these areas and show a brighter signal on contrast-enhanced MRI.

### 2.3. Physical Modeling of Flows in Microchannel Prototype “Y-Type Xylem Vascular Connection”

The study of flow hydrodynamics was carried out on an experimental stand (Figure 3a) using prototype “Y-type xylem vascular connection” (Figure 3d) with varying flow velocities over a wide range. The minimum velocity value was comparable to the flow of water through the xylem vessels in the maize stem (the matching approach is outlined in Section 2.4).

In the model experiment, the liquid (distilled water) flowed into one larger vessel from two smaller vessels at the velocities of $Q_{1}$ and $Q_{2}$, respectively. Several cases of $Q_{2}/Q_{1}$ ratio were considered: 0.25, 0.5, and 1. Such unequal inlet flows may simulate additional constrictions or perforated partitions, or selectively permeable membranes may be observed at the vessel junctions [7]. For definiteness, the flow velocity from the right channel was less than the flow velocity from the left channel, except for the case of equal flow rates in both channels. Note that in the experiment, for all considered ratios of input costs, their total consumption was equal, that is, $Q_{1}+Q_{2}=Q=const$.

With the value of Re=3, all considered cases of the ratios of the inlet flow velocities indicated laminar flow (Figure 4a). A decrease in the ratio of input costs led to a shift in the maximum velocity component to one of the channel walls, which may affect the efficiency of water transport from the xylem vessels to the other tissues of the shoot. The greatest change in the flow velocity was observed, when the ratio of input costs was $Q_{2}/Q_{1}=0.25$. In the initial section of the channel, the maximum value of the velocity increased by 0.4 compared with the flow velocity for $Q_{2}/Q_{1}=1$ and 0.5. At a distance of more than 800 µm, the flow velocity significantly decreased by 0.7.

Measurements were also made for the case of liquid flowing through the vessel at an average velocity of 0.25 mm per second (in dimensionless values, it corresponds to the number of Re=10; Figure 4b). The ratio of input rates also varied for the case of Re=3 but for Re=10; with unequal input rates, the velocity profile was extended along the channel, and the maximum value of the velocity increased by 0.25. This suggests that only due to the topology of the plexuses in the microchannel model, it is possible to obtain significant variations in the spatial distribution of water flow velocities and predict the hydraulic behavior inside a plant vascular system.

At a higher water velocity in the microchannel (1.2 mm per second; Re=47; Figure 4c), it was possible to identify the flow regions in which the velocity profile was significantly extended along the flow. For $Q_{2}/Q_{1}=1$, the maximum value, $2.5U/U_{0}$, of the flow velocity was achieved at a distance of 160 µm from the inlet channel. Next, downstream, the maximum flow velocity was decreased up to $2.2U/U_{0}$ at a distance of 640 µm from the inlet of the tube. The decrease in the value of the inlet flow ratio led to an increase in the maximum flow velocity in the channel input area to $3U/U_{0}$. At a distance of more than 800 µm, the velocity value and the velocity profile view become the same as for $Q_{2}/Q_{1}=1$. In the initial area of the tube, there was a transverse mass transfer, which, in turn, could contribute to the formation of vortices inside the vessel.

The study [34] also reported mixing the fluxes in a Y-junction, but they used an open microfluidic channel influenced by air. The obtained characteristics of the flow velocity are consistent with our experiment, considering the scale.

### 2.4. Reynolds-Number-Based Matching between Xylem Vessels in Maize Stem and Microchannels in Lab-on-a-Chip

We observed the flow hydrodynamics in prototype “Y-type xylem vascular bundles” for a wide range of velocities. On the other hand, for the model plant, we obtained LSM images for detailed structure reconstruction. In addition, during the acquisition of a series of MRI images, we fixed the volume of water that passed through the system of xylem vessels and evaporated through the leaves.

Since the plant grew under standard conditions in the growth chamber, and it was in the dark in the tomograph during the study, we can assume that the liquid entered into the xylem vessels evenly. Therefore, to estimate the average flow rate of fluid movement through the vessels in the section between two nodes, the following formula was used:(1)$$U_{0}=\frac{Q}{S},$$
where $U_{0}\left(m\;s^{-1}\right)$ is the mean flow velocity; $Q(m^{3}\;s^{-1})$ is the fluid flow through the plant, measured under experimental conditions; and $S(m^{2})$ is the cross-sectional area of all xylem vessels of the plant, providing the movement of water from the root to the leaves. The total cross-sectional area of all xylem vessels was calculated as a result of the analysis of LSM images for the transverse section of the stem and amounted to 358,292.39 µm${}^{2}$ (see the methods described in Section 4.2). Thus, the mean flow velocity of water through xylem vessels was $U_{0}=221.5$ µm s${}^{-1}$.

We used microfluidic chips to determine the features of fluid movement inside the vessels. The microfluidic chip configuration is described in Section 4.4. The microchannels in the chip are usually larger than the actual vessel in the plant, which allows one to use the modern methods of imaging diagnostics for flows with high spatial and temporal resolution [35] and to obtain the spatial quantitative characteristics of pressure, velocity, and diffusion of the flow inside the microchannel. The data obtained in the experiment for the spatial velocity distributions are shown in Figure 4.

The data presented in Figure 4 qualitatively show the features of fluid movement through the vessels. To quantitatively compare fluid motion in real xylem vessels and a model chip, we used the following approach: The Reynolds number (Re) is a similarity criterion in micro-, mini-, and macrodimensions and characterizes the ratio of the inertia forces acting on the flow to the viscous forces:(2)$$Re=\frac{U_{0}D_{h}}{\nu },$$
where $U_{0}$ ($m\;\;s^{-1}$) is the mean flow velocity; $D_{h}$ (m) is the characteristic vessel size; and ν ($m^{2}\;s^{-1}$) is the kinematic viscosity. In the experiment, distilled water with dissolved particles of gadolinium chelate was presented to the plant roots; the water temperature was 26 ${}^{\circ }$C, and the kinematic viscosity was $\nu =8.74\times 10^{-7}$
$m^{2}\;s^{-1}$. Since the cross section of xylem vessels is close in shape to ellipses (see Figure 5), then its hydraulic diameter $D_{h}$ could be taken as the characteristic size of the vessel:(3)$$D_{h}=\frac{4S}{P},$$

For the studied plant, the $D_{h}$ value of the vessels varied between 3.8 µm and 46 µm. Then, the Reynolds number calculated using formula (Equation 2) varied from 0.001 to 0.012 (Figure 5 and Appendix A).

Thus, the obtained values of the Reynolds number were quite low, and for a quantitative comparison of the velocity characteristics of the liquid inside the microfluidic chip and the plant vessels, it was worth considering the flows at equal Reynolds numbers.

## 3. Discussion

In this work, we demonstrated the use of horizontal tomograph Biospec 117/16 USR with a magnetic field strength of 11.7 T as a tool that allowed us to visualize the maize vascular system with high accuracy. The resulting images showed the complex topology of the connections of xylem bundles, which is consistent with data obtained using other methods [7,8]. The basis for nuclear magnetic resonance imaging is based on the following principles: Water molecules contain two protons. In a strong magnetic field, the alignment of the magnetic moments of the spin-bearing elements leads to a weak magnetization of the sample, which we can manipulate by applying time-dependent magnetic field pulses at the appropriate frequency (radio or high-frequency pulses). The magnetization component of the sample perpendicular to the main magnetic field induces a slight inductive voltage in the detector coil of radius *r* placed around the sample. The frequency components of the time-dependent inductive signal can be analyzed using a Fourier transform, resulting in a frequency spectrum. In a uniform magnetic field $B_{0}$, identical spins (for example, protons of water molecules) have the same Larmor precession frequency or resonant frequency, and a single resonant line is observed in the frequency spectrum. Inside the magnet, a well-defined constant magnetic field gradient *G* is created, with $G=\frac{\partial B_{0}}{\partial r}$, so identical spins at different positions along this gradient have different resonant frequencies, because the resonant frequency is proportional to the local magnetic field experienced. The spins can be oriented in three independent *x*-, *y*-, and *z*-directions or their combinations; therefore, they can be uniquely spatially encoded. Spin position mapping by magnetic field gradients can be performed in various ways [36]. Depending on the radiofrequency sequence used to excite the spins, the signal intensity in MRI images depends to a greater or lesser extent on the combination of parameters that reflect the spin density: relaxation time $T_{1}$ (spin–lattice) or $T_{2}$ (spin–spin). In addition to these parameters, the diffusion behavior of the molecules can also contribute to the contrast [37]. Regarding the use of MRI to determine the flow of water and solutions in the vascular system of a plant, it should be noted that in publications known to us, MRI was used to determine the flow of water in the vascular system of plant stems [38,39]. This is because of the fairly high speed of the convective flow in the stem vessels, which determines the successful application of MRI.

The contrast-enhanced MRI demonstrated in the current study is a promising tool for obtaining data on water content inside plant tissues with high time and spatial resolution. Such data could be used to discriminate between the convective and diffusion components of flows in a framework of models of plant tissue hydraulics [40] to clarify the mechanisms of the water pathways [41].

A brief discussion is warranted regarding the differences between natural xylem bundles and our microfluidic model. We studied the features of the movement of the fluid necessary for growth, nutrition, and photosynthesis, which occurs upward inside the plant from roots to leaves along the complex structure of xylem bundles, by assessing the hydrodynamics of the flow within the microfluidic system. In a system that implemented the so-called lab-on-a-chip approach, which models the connecting element of a conducting system consisting of alternating blocks of transverse nodal plexuses and longitudinal axial filaments, the inlet flows were regulated and varied (Figure 3a). Experimental systems designed to simulate transport processes in plants have been developed by several authors [29,42,43].

Plant xylem bundles play an important role as interfaces among roots, stems, and leaves. The description of the features of the movement of liquids through the vascular bundles after the nodes, depending on the input parameters, greatly contributes to the understanding of the transport of assimilates and minerals into plants. Non-contact field measurement methods with micron resolution are best suited for studying these processes. One of the most modern and effective methods for studying hydrodynamics in a microfluidic chip is micro Particle Image Velocimetry (micro-PIV). The method was proposed in 1998 by J.G. Santiago and C.D. Meinhart based on the PIV method for studying instantaneous and average velocity fields in liquids, gases, and flames. This method is based on the calculation of the movement of particles (tracers) in the flow section for a certain time interval. The micro-PIV method makes it possible to measure velocity vector fields in channels with micron resolution, while retaining the main advantages of the PIV method, such as non-contact and wide dynamic range. Hoffmann et al. pioneered these methods to study liquid–liquid flows in a T-channel by performing imaging and quantifying mixing efficiency by defining a concentration field [44]. The improvement of the measurement method was achieved in the works [45,46]. However, for many years, the error in determining the speed characteristics remained quite high. This problem was overcome in the work [35] and was also applied in this study. To calculate the velocity vectors, cells extended along the flow were used, which made it possible to achieve a relatively large dynamic range (the span between the maximum and minimum velocities). The inaccuracy of the offset determination did not exceed 0.1 pixel. Thus, the error of the method for measuring the average flow rate decreased to 1%.

The laminar flow of a liquid in a tube is known to be observed at low flow rates; in other words, Poiseuille flow occurs. The velocity profile becomes elongated along the flow direction and is approximated by a polynomial of the second degree. The double lengthening of the polynomial in the case where the velocity maximum is observed at values close to $2U_{0}$ (see Figure 4) is a feature of ultra-slow flows that are typically found in plant vessels. At the same time, small changes in the velocity profile with a different inflow of liquid into the tube can be observed. So, if the flow moves evenly across the entire width, then the velocity profile is symmetrical with respect to the center of the channel, and the maximum value of the velocity is reached in its central part. When the liquid enters the channel unevenly, the velocity profile shifts to one of the channel walls, and as a rule, an increase in the maximum velocity value in the initial section of the microchannel is observed. After the liquid passes about 1 mm of the path, the velocity profile becomes symmetrical about the central part of the channel again. However, in the plant vascular system, there are regions with more frequent connections than those necessary for hydraulic signal fading. Such mutual influence of fluxes could modify the behavior of the entire system and underlay the mechanism of self-regulation.

Thus, the use of lab-on-a-chip methods, together with the possibility of non-invasive measurement methods, can provide new frontiers in the study of the microscale movement of fluid through plant vessels. With the ability to vary and maintain flow inputs such as pressure and velocity and external factors such as ambient temperature and moisture content, the lab-on-a-chip can operate at steady state for many hours, allowing accurate measurements of dynamic processes within the system to be obtained.

## 4. Materials and Methods

### 4.1. Plant Material

Maize (*Zea mays* L.) variety B73 was used as an experimental plant for testing visualization tools. Plant samples were hydroponically grown in an environment-controlled growth chamber at 26 ${}^{\circ }$C and 70% relative humidity with 12 h of daily illumination (600 μmol m${}^{-2}$ s${}^{-1}$).

### 4.2. Laser Scanning Microscopy (LSM)

The studied stem samples were prepared for microscopy according to the following procedure: We incubated the tissues in fixative solution 3:1 (ethanol:acetic acid) for three or more hours. The section of the stem between the two nodes was sliced into 200 µm sections. Staining was performed with fluorescent dyes 4’,6-diamidino-2-phenylindole (DAPI; Sigma-Aldrich, Burlington, MA, USA) and Propidium Iodide (PI; Sigma-Aldrich). This set of dyes on a fixed material provides a fairly fade-resistant staining of nuclei and cell walls, including the vessels of the conducting system. Staining was carried out in two stages, with intermediate washing with neutral phosphate buffer. Leaf fragments washed twice were placed in 10 µg/mL PI solution for 30 min and then washed again twice for 15 min. Next, the fragments were stained with 10 µg/mL DAPI solution for 30 min and placed under a coverslip. DABCO reagent (Sigma-Aldrich) was used as an antifade reagent, which made it possible to carry out subsequent prolonged microscopic analyses. DABCO was diluted according to the standard procedure (25 mg/mL DABCO in 90% glycerol, 10% 1 × PBS, pH = 8.6). This mounting solution had sufficient viscosity to hold the coverslip on the sample fragment.

The stained samples of stem cross sections were imaged with an LSM 780 NLO microscope (Zeiss, Oberkochen, Germany) using objective Plan-Apochromat 20x/0.8 M27 in “tile scan” mode with 6 µm distance between adjacent optical sections along the vertical axis. The signal amplification was chosen dynamically. The pixel size in the optical sections was 0.692 × 0.692 µm. As a result, we obtained 2-channel 3D images for an extended region containing the whole stem in high resolution.

The tiles of 3D images were assembled into a single 3D-image using plugin LSM-W${}^{2}$ (ICG SB RAS, Novosibirsk, Russia, https://imagej.net/plugins/lsm-worker, accessed on 23 April 2022; [47]); then, the built-in functions of Fiji (https://fiji.sc/, accessed on 23 April 2022) for the maximum intensity projection of both channels converted the image to 2D.

Segmentation for the evaluation of the structural characteristics of the vascular system was performed manually by an expert. In the separate channel, xylem bundles and individual vessels were highlighted. Using built-in functions of Fiji (https://fiji.sc/, accessed on 23 April 2022), Image > Color > Split Channels, as well as functions of MorphoLibJ (http://imagej.net/MorphoLibJ, accessed on 23 April 2022, [48]), Plugins > MorphoLibJ > Segmentation > Morphological Segmentation, we obtained the data on dimensional and spatial distribution characteristics of individual vessels (Appendix A).

### 4.3. Magnetic Resonance Imaging

Images were gained on horizontal tomograph Biospec 117/16 USR (Bruker BioSpin, Ettlingen, Germany) with a magnetic field strength of 11.7 T.

As a contrast agent, Gadolinium-DTPA-BMA (Gadolinium 5,8-bis(carboxylatomethyl)-11-[2-(methylamino)-2-oxoethyl]-3-oxo-2,5,8,11-tetraazatridecan-13-oate) [49] was used. Plant roots were exposed in a 5 mM solution. The studied area was in the bottom part of the maize shoot, starting from the junction between the root and stem, and was 18 mm long. The area was scanned in two successive 9 mm series. As a reference, a test tube with distilled water was placed next to the plant. The distance between slices was 1 mm, and each slice averaged information about 750 µm along the growth axis. This area was scanned by equal intervals over 7 h. The studied plant was not removed from the tomograph until the end of the experiment. Thus, we obtained a spatio-temporal series of slices for a maize stem. All the images were exported as dicom files in ParaVision 5.1 Software (Bruker, Ettlingen, Germany).

On each image, areas corresponding to a stem, a test tube with water, and air were segmented. For each of these segments, the mean pixel intensity values were calculated. To consider the noise in the data, the final value that determined the flow of water with gadolinium was calculated using the following formula:(4)$$S\left(flow_{i}^{t}\right)=\frac{S\left(stem_{i}^{t}\right)-S\left(stem_{i}^{0}\right)}{S(air_{i}^{t})},$$
where $X_{i}^{t}$ is the region on image i∈{1,…,18} in time moment t∈{1,…,19}, and $S(X_{i}^{t})$ is the mean value of pixel intensity in area $X_{i}^{t}$. The values according to Formula (Equation 4) were calculated for each slice at each time point. The resulting distribution is shown in Figure 2d.

### 4.4. Particle Image Velocimetry

An experimental study of the hydrodynamic characteristics of fluid flow in the microchannels was carried out in a unique scientific facility at Institute of Thermophysics of the Siberian Branch of the Russian Academy of Sciences. The experimental stand included a syringe pump (Gemini 88 Plus Dual Rate Syringe Pump; KD Scientific, Billerica, MA, USA), a working area, a microscope (Carl Zeiss Axio Observer.Z1; Oberkochen, Germany), and a measuring system for Particle Image Velocimetry (Novosibirsk, Russia). The work [50] contains a detailed description of the facility design.

The measurements of the velocity characteristics of the flow were carried out using a microfluidic chip with two inlets connected to each other at an angle of 60 degrees and one outlet (Figure 3d). The inlet channels had a square section with a side equal to 120 μm, and the outlet channel had a rectangular section of 120 × 240 μm. The aspect ratio was 1:1:2. The length of the inlet sections was 0.7 cm, and the length of the outlet section was 2 cm. The channel walls were made of optically transparent material SU-8 (microLIQUID, Mondragon, Spain). Irregularities on the channel walls did not exceed 20 μm. The syringe pump provided a continuous supply of fluid to the microchannel inlets. The fluid flow rate at the inlets varied from 0.39 mL/h to 115 mL/h. Distilled water was used in the experiment as the working fluid. The Reynolds number (Re) varied from 3 to 100.

For definiteness, we assumed that the input flow rate in the right channel was less than or equal to the flow rate in the left one ($Q_{2}\leq Q_{1}$), and their sum was constant in all cases ($Q_{1}+Q_{2}=Q$). The ratio of input costs was expressed by coefficient $R=Q_{2}/Q_{1}$.

The velocity characteristics of the currents were measured using the Particle Image Velocimetry (PIV) method with micron resolution, using a double-pulsed Neodymium:Yttrium-Aluminum-Garnet laser (radiation wavelength, 532 nm; pulse repetition rate, 4 Hz; pulse duration, 10 ns; pulse energy, 25 mJ), a CCD camera (depth, 8 bits; matrix resolution, 2048 × 2048 pixels), and a synchronizing processor (Figure 3a). The spatial resolution was 1 µm per pixel. The measuring system was controlled by software package ActualFlow (Russia, Novosibirsk, http://polis-instruments.ru/, accessed on 23 April 2022). To carry out PIV measurements, 2 μm fluorescent tracer particles (red fluorescent polymer microspheres; Rhodamine 6G; Thermo Fisher Scientific, Waltham, MA, USA) were added to the flow (Figure 3c).

We obtained 5000 images for each flow regime, which made it possible to minimize the measurement error of the velocity fields [51]. An iterative cross-correlation algorithm with continuous displacement and deformation of elementary computational domains and 50% overlap between them was used to calculate velocity fields. The sub-pixel interpolation of the cross-correlation peak was carried out on three points, using one-dimensional Gaussian approximation. In order to have a relatively large dynamic range, the size of the initial computational area corresponded to 128 × 32 pixels. The size of the final computational area was 32 × 8 pixels to provide a relatively high spatial resolution. The error for determining the offset did not exceed 0.1 pixel. Thus, when the tracers were shifted by 8 and 2 pixels, the velocity measurement errors were 1% and 4%, respectively. The validation of the calculated velocity vectors took place in two stages: validation by the signal-to-noise ratio with a threshold of 2 and adaptive median filtering with a region size of 7 × 7. An example of the obtained instantaneous velocity field is shown in Figure 3e. At the next step, we averaged the instantaneous velocity fields and obtained the field of the average current velocity of the measurement area. Based on the average current velocity fields, the velocity profiles were constructed.

## 5. Conclusions

Modern imaging techniques make it possible to study plants’ functioning features at different levels, from the ultrastructure of cells and tissues to the organization of plant communities. Systems biology approaches allow one to integrate these levels into complete model water flows through the soil–plant–atmosphere continuum. The most significant outcomes can be obtained as a result of the assembling of approaches at different levels. This paper shows a way to combine imaging tools for water flow visualization inside the maize stem vascular system, including contrast-enhanced MRI, laser scanning microscopy, and the microfluidic-chip modeling of an element of vessel connection. In the obtained images, the pixel sizes differed by two orders of magnitude. We believe that new peculiarities of plant vascular system functioning could emerge from such a comprehensive and quantitative consideration of the various determinants of water flow within the plant vascular system.

## Figures and Tables

**Figure 1 plants-11-01533-f001:**
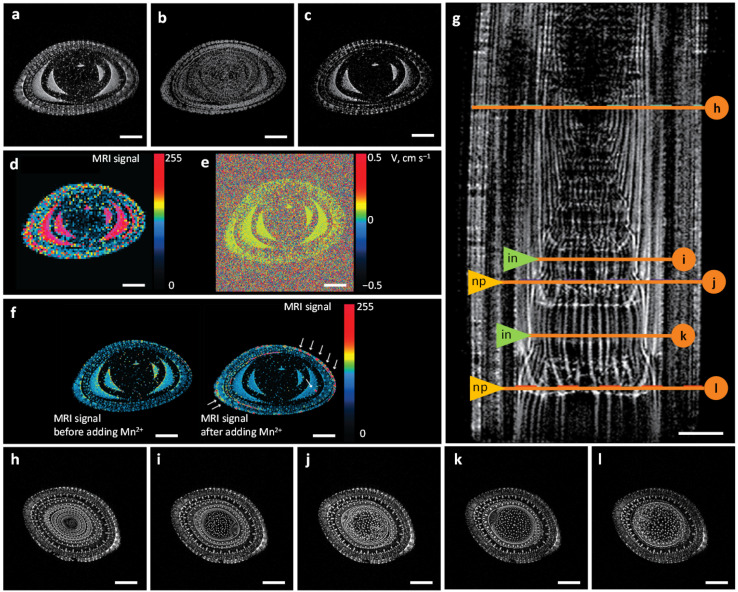
MRI methods for mapping the water distribution in maize stem tissues: $T_{2}$-weighted MRI (**a**); proton-weighted maps of non-aqueous (**b**) and aqueous (**c**) proton distributions; diffusion-weighted MRI (**d**) and velocity map of water flow through a virtual slice (**e**). (**f**) $T_{1}$-weighted MRI before and after root exposure to MnCl${}_{2}$ solution (10 mM); the arrows indicate the locations of manganese ion accumulation; pseudo-staining reflects the intensity distribution of the MRI signal on the scan; MRI scans are axially oriented. (**g**) $T_{2}$-weighted MRI image of the growth zone of leaves and stem of maize; a longitudinal section of the stem (**g**) and some transverse sections (**h**–**l**) located in internodes (in, in the figure) (for (**i**,**k**)) and nodal plexuses (np, in the figure) (for (**j**,**l**)) are indicated; the pixel size was 56 × 56 µm, and the slice thickness was 0.5 mm. Turbo Rapid Imaging with Refocused Echoes (TurboRARE) method.

**Figure 2 plants-11-01533-f002:**
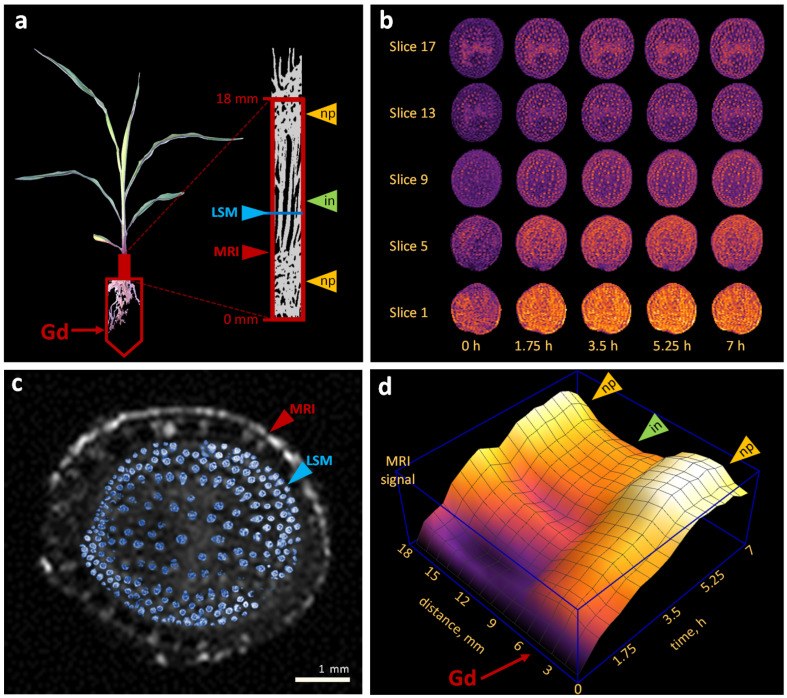
Contrast-enhanced MRI study of gadolinium chelate distributing inside nodal plexuses (nps) and internodes (ins) of the maize stem. (**a**) The MRI-studied area marked by the red frame was at the bottom of the maize stem, from the root/shoot nodal plexus (np) to a height of 18 mm. The location of the internode (in) section for high-resolution LSM-imaging is marked with a blue arrow. (**b**) Temporal series of MRI slices. (**c**) Matching of internodal vascular bundles in the MRI slice and in the high-resolution LSM-image. (**d**) Space–time distribution of MRI signal reflecting Gd penetration into the maize stem xylem vascular system. The red arrow indicates the roots/stem connection region, which can be interpreted as a source of gadolinium chelate.

**Figure 3 plants-11-01533-f003:**
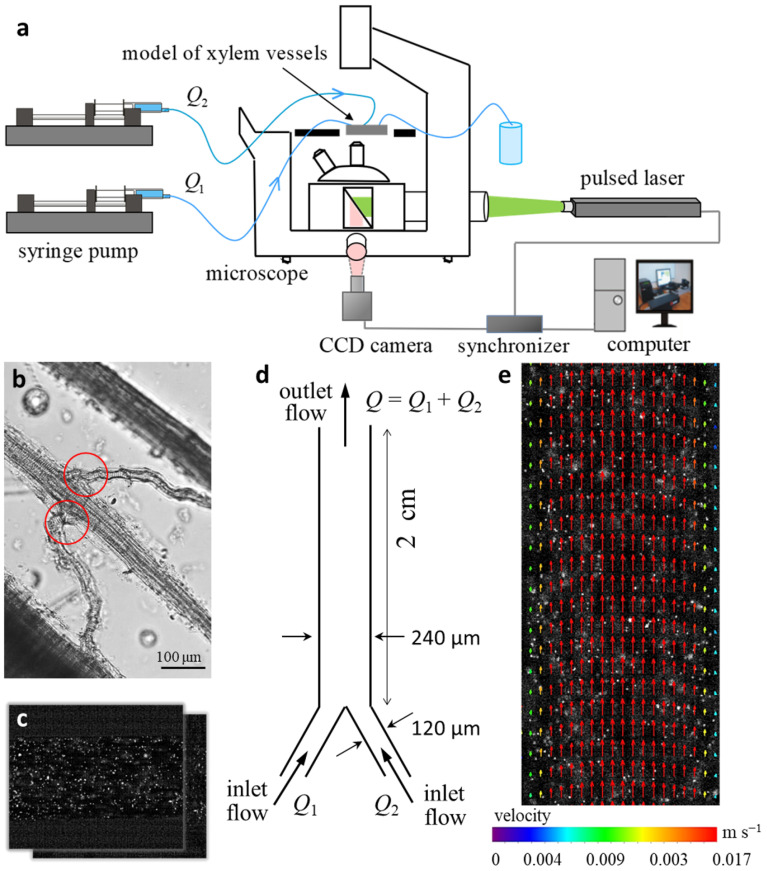
Particle Image Velocimetry. (**a**) General scheme of the experimental stand. (**b**) An example of xylem vessel connections (marked by red circles) corresponding to microchannel prototype “Y-type xylem vascular connection”. The image of xylem vessels in a maize leaf was obtained after 2 h of enzymatic degradation of tissues under bright light on an Olympus BX53 microscope (Olympus corporation, Japan) with 20× magnification. (**c**) Double image of tracer particles in model microchannel. (**d**) Scheme of prototype “Y-type xylem vascular connection”. (**e**) Instantaneous velocity field obtained for the double image superimposed on the tracer particle image.

**Figure 4 plants-11-01533-f004:**
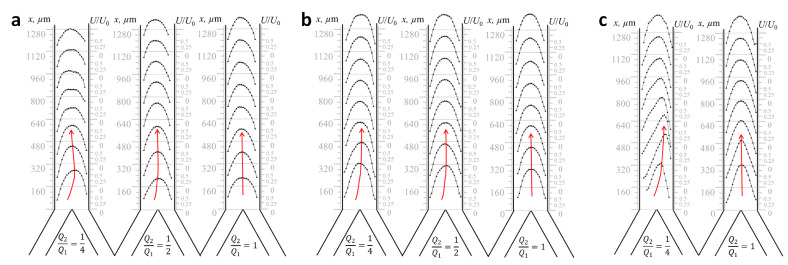
Longitudinal component of the average flow velocity in prototype “Y-type xylem vascular connection” resulted from different ratios of inlet flow rates and equal total flow rate in the outlet channel. (**a**) Re=3 equaled to 0.1 mm per second; (**b**) Re=10 equaled to 0.25 mm per second; (**c**) Re=47 equaled to 1.2 mm per second.

**Figure 5 plants-11-01533-f005:**
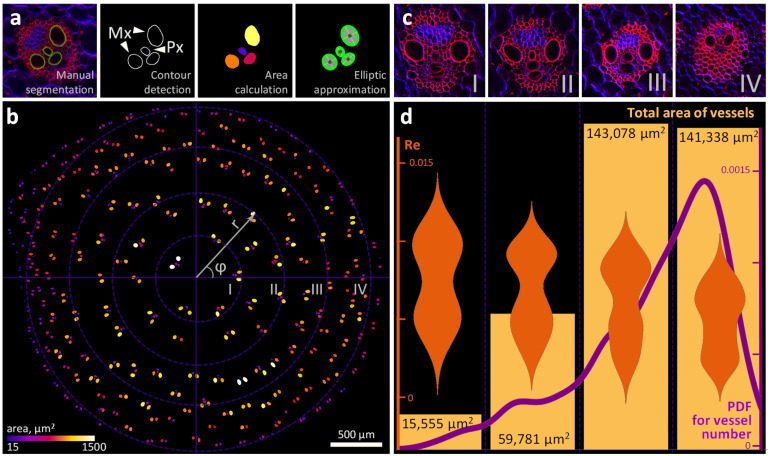
LSM image-based determination of numerical characteristics for Reynolds number-based matching between the xylem vessels in the maize stem and the microchannels in the lab-on-a-chip model. (**a**) The main stages of LSM image processing for obtaining data on the spatial distribution of the dimensional characteristics of the xylem vessels (metaxylem (Mx) and protoxylem (Px)) in the internode of the maize stem, including expert manual segmentation, contour extraction, calculation of cross-sectional areas, and ellipse fitting. (**b**) The location of the xylem vessels on the cross section of the maize stem in the I, II, III, or IV “rings”; the coordinates were recalculated into the polar system relative to the nominal center of the stem; the color indicates the cross section of the vessel area. (**c**) LSM images for typical xylem bundles located in the I, II, III, and IV “rings”. (**d**) For xylem bundles located in the I, II, III, and IV “rings”, the distribution of vessels relative to the center of the stem, the total cross-sectional areas, and violin plots for Reynolds numbers (Re).

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
