# Peer review of "Particle-Based Imaging Tools Revealing Water Flows in Maize Nodal Vascular Plexus"

_plants, 2022, doi:10.3390/plants11121533_

Round 1

Reviewer 1 Report

This is an interesting paper on the flow of sap across notes and internodes of maize plants. The authors apply novel methods, and generally present the data in an accurate and clear way. I was pleased to see the Particle Image Velocimetry and the Reynolds numbers based on the flow simulations, as this topic has been poorly addressed in the past. Therefore, the paper provides a welcoming contribution and opens up novel perspectives to study flow. This is especially important at nodes, as plants can be highly compartmentalised or segmented. The English style and grammar is fairly good, but needs to minor editing.

I have only few comments:

Line 29-30: is there a reference to make the claim that only 3% of axial vessels pass through nodes straight?

Figure caption 1: add the meaning of in (internodes) and np (nodal plexes)

Line 102: shown

Figure 3b: I am not sure if this is really a Y-shaped vessel as shown in Fig. 3d. There is a main vein, and two side veins connected to it. However, is the connection via intervessel pits, or is the vessel directly bifurcating as suggested? This cannot be seen on the image. Vessels in nodes can be highly variable, with circular to zigzagging patterns. See for instance the work by Jean-Pierre André, and some of the references cited. Still, it is fine to have a Y-shaped vessel as an exercise for the PIV method.

Author Response

Dear reviewer, thank you for your detailed review of our work. We have modified our manuscript according to your suggestions.

Point 1: Line 29-30: is there a reference to make the claim that only 3% of axial vessels pass through nodes straight?

Response 1: This data are from the work of Shane et al., the reference was indicated in the appropriate place.

Shane, M. W., McCully, M. E., & Canny, M. J. (2000). The vascular system of maize stems revisited: implications for water transport and xylem safety. Annals of Botany, 86(2), 245-258.

Point 2: Figure caption 1: add the meaning of in (internodes) and np (nodal plexes)

Response 2: We corrected the Figure caption 1.

Point 3: Line 102: shown

Response 3: We checked the text.

Point 4: Figure 3b: I am not sure if this is really a Y-shaped vessel as shown in Fig. 3d. There is a main vein, and two side veins connected to it. However, is the connection via intervessel pits, or is the vessel directly bifurcating as suggested? This cannot be seen on the image. Vessels in nodes can be highly variable, with circular to zigzagging patterns. See for instance the work by Jean-Pierre André, and some of the references cited. Still, it is fine to have a Y-shaped vessel as an exercise for the PIV method.

Response 4: The vascular system of plants has a complex structure and various types of connections between vessels. The inner structure of nodes and internodes complicate this structure in grasses, in particular in maize (Kraehmer, 2017). We were not goaled to make a universal model for vascular connections, but mainly to show the fundamental possibility of such a study.

In the leaf of cereal plants, there are longitudinal and transverse xylem vessels, which are connected by 3-way junctions. Figure 3b shows two such connections between two parallel vessels in maize leaf.

It is important to note that these connections correspond to the microfluidic model (Figure 3d) only in topology. In the same time in a living systems, additional constrictions, perforated partitions or selectively permeable membranesare observed at the vessel junctions (Shane et al., 2000). Such structures can be modeled by changing the velocities for inlet flows (some cases are demonstraned in Figure 4). Therefore, the model proposed by us requires further implementation to a wide range of cases.

We have changed the caption for Figure 3b:

“An example of xylem vessels connections (marked by red circules) corresponding to microchannel prototype "Y-type xylem vascular connection". The image of xylem vessels in a maize leaf was obtained after 2 hours of enzymatic degradation of tissues in bright light on Olympus BX53 microscope (Olympus corporation, Japan) with 20x magnification.”

and added an appropriate explanation to the text of the Results (Lines 162-163).

Also we added a reference to Jean-Pierre André's fundamental work.

Kraehmer, H. (2017). On vascular bundle modifications in nodes and internodes of selected grass species. Scientia agriculturae bohemica48(3), 112-121.

Shane, M. W., McCully, M. E., & Canny, M. J. (2000). The vascular system of maize stems revisited: implications for water transport and xylem safety. Annals of Botany, 86(2), 245-258.

Reviewer 2 Report

The manuscript has some promise but there are a few major problems that need to be addressed:
- Introduction does not make clear what are the main research challenges that still need to be properly addressed, how the proposed method improves upon its predecessors, nor what are the main contributions of the study. Without proper contextualization, it is difficult to evaluate the significance of the research.
- It would be better if the Materials and Methods section was located just after the introduction.
- Results are presented in a qualitative and subjective way, without any comparison with some reference or other methods. A more objective way to evaluate the technology is needed for proper evaluation of the results.

Author Response

Dear reviewer, thank you for your detailed review of our work. We have modified our manuscript according to your suggestions.

Point 1: Introduction does not make clear what are the main research challenges that still need to be properly addressed, how the proposed method improves upon its predecessors, nor what are the main contributions of the study. Without proper contextualization, it is difficult to evaluate the significance of the research.

Response 1: We have significantly revised the Introduction. Thank you for your advice.

Point 2: It would be better if the Materials and Methods section was located just after the introduction.

Response 2: We have rechecked, according to “Instructions for Authors” (https://www.mdpi.com/journal/plants/instructions), the following structure is required:

Research manuscript sections: Introduction, Results, Discussion, Materials and Methods, Conclusions (optional).

We prepared the manuscript according to these instructions. Therefore, we would refrain from rearranging parts of the manuscript. I think we can discuss this matter with the editor of the special issue.

Point 3: Results are presented in a qualitative and subjective way, without any comparison with some reference or other methods. A more objective way to evaluate the technology is needed for proper evaluation of the results.

Response 3: Thanks for the note, we revised the text in Results. The most significant changes are in Lines 153-158, 177-183, and 194-204.

Round 2

Reviewer 2 Report

My comments have been properly addressed.